# Harnessing clinical annotations to improve deep learning performance in prostate segmentation

**Karthik V. Sarma** [1], **Alex G. Raman** [1,2], **Nikhil J. Dhinagar** [1,3], **Alan M. Priester** [1], **Stephanie Harmon** [4,5], **Thomas Sanford** [4,6], **Sherif Mehralivand** [4], **Baris Turkbey** [4], **Leonard S. Marks** [1], **Steven S. Raman** [1], **William Speier** [1], **Corey W. Arnold** [1] *

1 University of California, Los Angeles, Los Angeles, CA, United States of America, 2 Western University of Health Sciences, Pomona, CA, United States of America, 3 Keck School of Medicine, University of Southern California, Los Angeles, CA, United States of America, 4 National Cancer Institute, National Institutes of Health, Bethesda, MD, United States of America, 5 Clinical Research Directorate, Frederick National Laboratory for Cancer Research, Frederick, MD, United States of America, 6 SUNY Upstate Medical Center, Syracuse, NY, United States of America

* cwarnold@ucla.edu

**Data Availability Statement:** The underlying private clinical imaging data cannot be shared publicly because of patient privacy and compliance requirements. The PROMISE12 grand challenge

## Abstract

### Purpose

Developing large-scale datasets with research-quality annotations is challenging due to the high cost of refining clinically generated markup into high precision annotations. We evaluated the direct use of a large dataset with only clinically generated annotations in development of high-performance segmentation models for small research-quality challenge datasets.

### Materials and methods

We used a large retrospective dataset from our institution comprised of 1,620 clinically generated segmentations, and two challenge datasets (PROMISE12: 50 patients, ProstateX-2: 99 patients). We trained a 3D U-Net convolutional neural network (CNN) segmentation model using our entire dataset, and used that model as a template to train models on the challenge datasets. We also trained versions of the template model using ablated proportions of our dataset, and evaluated the relative benefit of those templates for the final models. Finally, we trained a version of the template model using an out-of-domain brain cancer dataset, and evaluated the relevant benefit of that template for the final models. We used five-fold cross-validation (CV) for all training and evaluation across our entire dataset.

### Results

Our model achieves state-of-the-art performance on our large dataset (mean overall Dice 0.916, average Hausdorff distance 0.135 across CV folds). Using this model as a pre-trained template for refining on two external datasets significantly enhanced performance (30% and 49% enhancement in Dice scores respectively). Mean overall Dice and mean average Hausdorff distance were 0.912 and 0.15 for the ProstateX-2 dataset, and 0.852

dataset is made available by its owners here: https://promise12.grand-challenge.org/ and the ProstateX-2 grant challenge dataset is made available by its owners here: https://www.aapm.org/GrandChallenge/PROSTATEx-2. The values behind the means, standard deviations and statistical tests reported are available on the Dryad repository with the DOI 10.5068/D1J09F.

**Funding:** KVS acknowledges support from National Cancer Institute grant F30CA210329, National Institute of General Medical Studies grant GM08042, and the UCLA-Caltech Medical Scientist Training Program. CWA acknowledges funding from National Cancer Institute grants R21CA220352 P50CA092131. LSM acknowledges funding from National Cancer Institute grants R01CA195505 and R01CA158627. SH acknowledges that this project has been funded in whole or in part with federal funds from the National Cancer Institute, National Institutes of Health, under Contract No. HHSN261200800001E. TS, SM, and BT acknowledge that this project was supported in part by the Intramural Research Program of the NIH. The content of this publication does not necessarily reflect the views or policies of the Department of Health and Human Services, nor does mention of trade names, commercial products, or organizations imply endorsement by the US Government. The funders had no role in study design, data collection and analysis, decision to publish, or preparation of the manuscript. There was no additional external funding received for this study.

**Competing interests:** LSM and AMP report a financial interest in Avenda Health outside the submitted work. BT reports IP-related royalties from Philips outside the submitted work. The NIH has cooperative research and development agreements with NVIDIA, Philips, Siemens, Xact Robotics, Celsion Corp, and Boston Scientific outside the submitted work. The NIH has research partnerships with Angiodynamics, ArciTrax, and Exact Imaging outside the submitted work. CWA has received research equipment from NVIDIA Corporation, outside the submitted work. No commercial funding or equipment was used in the execution of this study. No other authors have competing interests to disclose.

and 0.581 for the PROMISE12 dataset. Using even small quantities of data to train the template enhanced performance, with significant improvements using 5% or more of the data.

## Conclusion

We trained a state-of-the-art model using unrefined clinical prostate annotations and found that its use as a template model significantly improved performance in other prostate segmentation tasks, even when trained with only 5% of the original dataset.

## Introduction

Prostate cancer is the second most frequent cancer diagnosis and the fifth leading cause of death for men worldwide [1]. Prostate segmentation is a component of the routine evaluation of prostate magnetic resonance imaging (MRI) necessary both for surveillance (through volume estimation) as well as targeted biopsy (to enable registration with real-time ultrasound). In the segmentation workflow, a clinician (generally a radiologist or urologist) will manually review the slices of a 3D T2-weighted MR image and produce a contour for each slice. In some workflows, the radiologist will use a computer-assistance tool, such as DynaCAD Prostate (Invivo-Philips, Gainesville, Florida) [2], to assist in segmentation, either by first producing an approximate annotation that is then edited by the radiologist, or by providing an assisted drawing tool that heuristically supports the designation of a contour. Regardless of workflow, segmentation requires a slice-by-slice analysis, which is time consuming, requires the skills of a specially trained radiologist, and is prone to intra- and inter-reader variability [3]. In addition to the utility of such segmentations for these clinical applications, obtaining a precise segmentation is critical for supporting image analysis research, as incorrectly assigning image regions may impair trained classifier accuracy, particularly in the case of lesion detecting classifiers that rely on input prostate segmentations as a component of the input pathway.

Automated prostate segmentation is an active area of research, and substantial published work exists on the development of machine learning models for the purpose. However, these state of the art prostate segmentation algorithms [4–9] are often trained on small research-quality annotated datasets curated specifically for machine learning. Examples include the 100 patient Prostate MR Image Segmentation (PROMISE12) challenge dataset [10] and the 60 patient NCI-ISBI (National Cancer Institute–International Symposium on Biomedical Imaging) Automated Segmentation of Prostate Structures (ASPS13) challenge dataset [11]. Other algorithms have been trained on institutionally developed local datasets that include between 100 and 650 studies [12–15]. Unfortunately, the development of research-quality prostate boundary annotations is challenging. For example, for the PROMISE12 dataset, segmentations were created by an experienced radiologist, verified by a second experienced radiologist, and then re-annotated by a third nonclinical observer—a complex and expensive process.

If, however, rough clinical annotations could be used to enable training a highly accurate segmentation model, these issues could be avoided, and substantially more data could be available. In this study, we train a prostate segmentation model using a large clinical prostate MRI dataset and rough clinical annotations created as part of the clinical workflow at our academic medical center. We then explore generalizing that model through refinement with small datasets, and the impact of original dataset size on generalizability. Finally, to confirm that it is the prostate specific features in our model that improve generalization rather than general MR features, we explore the relative utility of using our pretrained prostate model for as a basis for generalization versus a model pretrained on an MR dataset from brain cancer patients.

## Materials and methods

### Data

Four retrospective sources of data were used for this project. For training our segmentation model with our clinical data, we used MRI data collected from patients seen at our institution during routine clinical procedures. For examining generalization, we made use of two research-quality prostate MRI challenge datasets. Finally, for determining the relative utility of using our model trained with clinical data as a pre-trained starter, we made use of a brain MRI challenge dataset for comparison. All data was used for this work under the approval of the University of California, Los Angeles (UCLA) institutional review board (IRB# 16–001087). Informed consent was waived with the approval of the IRB for this retrospective study of medical records, based on institutional guidelines, the fact that the study involved no more than minimal risk, the fact that the waiver would not adversely affects the rights and welfare of study patients, and the impracticality of conducting the retrospective analysis in which results would not change care already delivered to study patients. Data used for this study was de-identified after collection and before analysis.

**Primary dataset.** Our internal clinical population for this study consists of 1,620 MRI studies collected from 1,111 patients who underwent transrectal ultrasound-MRI fusion biopsy (TRUS biopsy) using the Artemis guided biopsy system (Eigen Systems) between 2010 and 2018 at our institution using a standardized protocol and 3T magnet (Trio, Verio, or Skyra, Siemens Healthcare). As part of the protocol, prostate MRIs were contoured in a two-part process. First, the attending radiologist for the case (the attending radiologists for the patients included in this study each had between 10–27 years of experience) created a prostate contour using the DynaCAD Prostate image analysis platform as part of the routine clinical workflow. This contour was then used by a technician to re-contour the prostate on the Profuse (Eigen Systems) platform in order to enable use with the Artemis biopsy system, as DynaCAD segmentations cannot be directly imported for use on the Artemis.

We retrospectively collected 3D axial turbo spin echo (TSE) T2-weighted images and prostate contour sets from these studies. T2 images were acquired clinically using the spatial and chemical-shift encoded excitation (SPACE, Siemens Healthcare) protocol, with field of view (FOV) 170 x 170 x 90 mm$^3$ and resolution 0.66 x 0.66 x 1.5 mm$^3$. Acquisition parameters are provided in **S1 Table**. Studies were collected from our institution's picture archiving and communication system (PACS). Corresponding T2 prostate contours were collected from the Profuse image analysis platform. Imaging data were collected from every available study for each patient seen at our institution during the study period. Studies were excluded from retrieval if the T2 image or contour was missing from PACS or corrupt, or if the image exhibited a protocol deviation, such as a variance in FOV or resolution. A total of 1,620 studies were included from 1,111 patients, and 84 studies were excluded. Of the 1,620 included studies, 29 used an endorectal coil.

**External prostate challenge datasets.** Two external challenge prostate datasets were used for this study: ProstateX-2 [16] and PROMISE12 [10].

The ProstateX-2 Challenge was a prostate cancer prediction challenge held in 2017. This dataset consists of 99 deidentified cases collected from patients seen at Radboud University Medical Center in the Netherlands. A consistent imaging protocol was used for all cases, which was significantly different from the protocol used for the primary dataset at our institution. A variety of images and clinical variables were provided with each case. For use in our experiments, we retrieved transverse T2-weighted MR images from each case in the dataset. These images were then annotated with a research-quality prostate contour by co-author B.T., an experienced abdominal radiologist.

The PROMISE12 Grand Challenge was a prostate segmentation-specific challenge held in 2012. This dataset includes 50 deidentified cases collected from four different centers (Haukeland University Hospital in Norway, Beth Israel Deaconess Medical Center in the United States, University College London in the United Kingdom, and Radboud University Nijmegen Medical Center in the Netherlands). Each institution had unique acquisition protocols, with wide variability in the MR field strength, endorectal coil usage, and image resolution. Each case consists of a transverse T2 weighted MR image and a reference research-quality prostate contour produced by agreement of two expert radiologists (one radiologist at the institution where the image was acquired, and a second radiologist at Radboud University). Detailed acquisition parameters are not available for this dataset, but images were scanned at a variety of field strengths (1.5T or 3T), with or without endorectal coil, and with a variety of acquisition resolutions, pulse sequences, and device manufacturers [10].

**Brain cancer dataset.**   In order to provide a non-prostate comparison, the Brain Tumor Segmentation (BraTS) 2019 [17, 18] challenge dataset was also used for this study. The dataset includes over 300 annotated cases collected from 19 different institutions using a wide variety of protocols. These cases include T2-weighted images of the brain with tumor segmentations. These segmentations were created manually using a multi-step protocol requiring agreement between multiple raters and final approval by an experienced neuroradiologist. Though tumor segmentation is a far more complex segmentation task than organ segmentation, this dataset provided an MRI comparison with a defined 3D segmentation task that could be leveraged as pretraining for prostate MRI segmentation. The BraTS 2019 data originates from large number of institutions and includes data collected with a variety of acquisition parameters; a specific breakdown of these parameters is not available [19].

## Preprocessing

In order to facilitate transportability, we processed images from all three datasets using the same pipeline. Initial preprocessing was done in Python, primarily using the SimpleITK toolkit [20], and included bias field correction [21] and resampling to isotropic voxel size (1mm x 1mm x 1mm) for further processing; these steps were based on preprocessing done in previous work [13, 21–23]. After initial preprocessing, we applied interquartile range (IQR)-based intra-image normalization to address the relative nature of MR image intensity values (both within and between institutions). Each image was normalized to the image-level IQR (calculated from the central 128x128 column of the volume) and then values were clipped between two IQRs below the first quartile and five IQRs above the third quartile, in order to eliminate outlying values created by imaging artifacts. The preprocessing pipeline is depicted in **Fig 1**.

## Augmentation

For all model training in this study, real-time augmentation using the Batchgenerators package was performed [24]. Three augmentation transformations were used: 1) random elastic deformation, 2) random rotation in the range [-$\pi$/8, $\pi$/8] in the axial plane, and [- $\pi$/4, $\pi$/4] along the axis, and 3) random mirroring across the *y* axis. After augmentation, the image was cropped to the central column of the transformed image (i.e. the central 128x128 voxels in the *x,y* plane).

## Model, training and evaluation

The base model used for this study was the 3D U-Net [25]. For all experiments, the network was configured with four encoder levels, three decoder levels, a ReLU transfer function and group normalization (using eight groups) following every convolutional layer, and a softmax

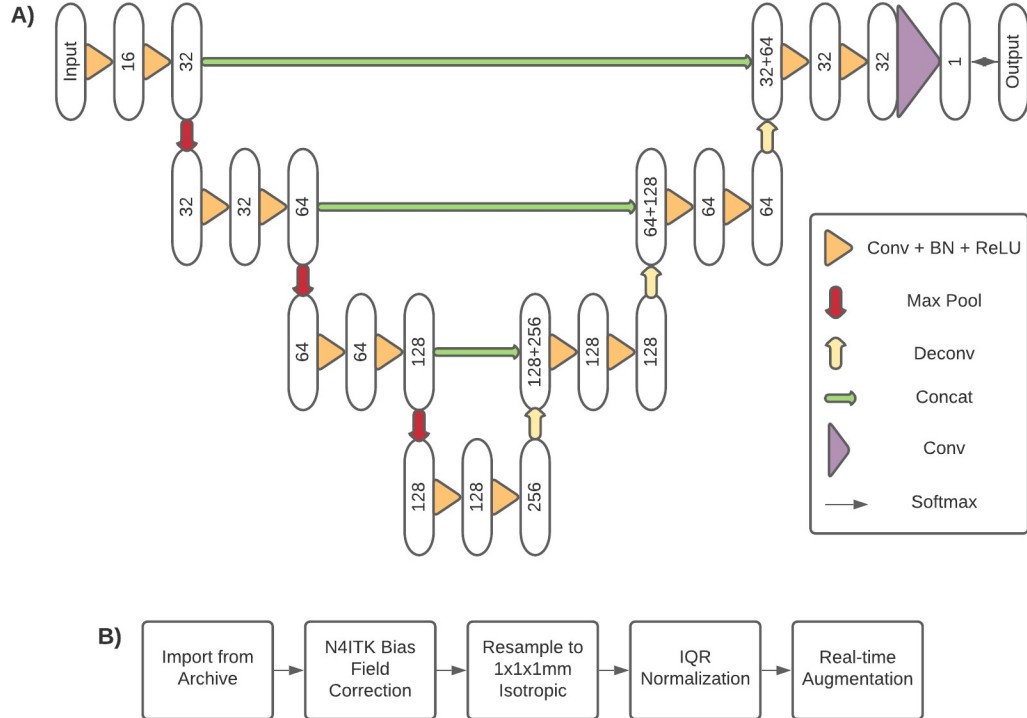

**Fig 1. 3D U-Net model diagram and preprocessing steps.** A) Network diagram of the 3D U-Net used for this study. Numbers within the ovals represent number of feature maps at that layer. Connections represent network operations, such as 3x3x3 3D convolution ("Conv"), 2x2x2 max pooling ("Max Pool"), 3x3x3 3D transposed convolution ("Deconv"), skip feature map concatenation ("Concat"), batch normalization ("BN"), rectified linear unit activation ("ReLU"), and softmax output ("Softmax"). B) Process diagram of preprocessing steps. Once images were imported from the archive (either PACS or challenge download), N4ITK bias field correction was applied. Images were then resampled to 1mm isotropic resolution and IQR normalized. During training, real-time augmentation was applied to each input image to create the training sample for that epoch.

output layer. The model architecture is depicted in **Fig 1**. All training and evaluation was done using the PyTorch framework on a DGX-1 (NVIDIA) deep learning appliance. Mixed-precision training using the NVIDIA Accelerated Mixed Precision (AMP) was used at optimization level O2, consisting of 16-bit model weights and inputs, 32-bit master weights and optimizer parameters, and dynamic loss scaling.

Network inputs consisted of the full augmented image volume (with dimension 128x128x136). Training was performed using the Adam optimizer with learning rate $10^{-5}$ and the soft Dice loss function. Each epoch consisted of training on a full dataset comprised of one augmented sample generated for every original input sample.

The primary evaluation metric used to compare segmented volumes was the soft Dice coefficient function as denoted in Eq 1, where $S_{DL}$ is the segmentation of a deep learning model and $S_m$ is the manual segmentation. The value of the coefficient can range between 0 (no overlap) and 1 (perfect overlap).

$$Dice\left(S_{DL}, S_m\right) = \frac{2|S_{DL} \cap S_m|}{|S_{DL}| + |S_m|} \tag{1}$$

The average Hausdorff distance (AHD) was also used as a secondary metric, as denoted in Eq 2, where $X$ is the set of all points within the manual segmentation, $Y$ is the set of all points within the segmentation of the deep learning model, and $d$ is the Euclidean distance. The

AHD is a positive real number, and smaller numbers denote better matching segmentations.

$$dd_{AHD}(X, Y) = \frac{\frac{1}{|X|}\sum_X \min_Y \ d(x, y) + \frac{1}{|Y|}\sum_Y \min_X \ d(x, y)}{2} \quad (2)$$

The evaluation metrics were calculated for whole prostate gland segmentation on the entire uncropped volume. In addition, each slice of the segmentation mask was split into three sub-volumes: the apex subvolume (consisting of the apical 25% of prostate slices), the base subvolume (consisting of the basal 25% of prostate slices), and the midgland subvolume (consisting of the remaining middle 50% of slices); the Dice evaluation metric was calculated for each sub-volume. Means and standard deviations across the entire dataset were reported for performance on the whole prostate as well as each of the three subvolumes. These were calculated using the following approach: for each of the five folds, metrics were calculated for each of the images within the fold using the model trained without that fold's data. Once the metrics were calculated for every study, the mean and standard deviation of each metric across all images was computed (including whole-volume and subvolume metrics) and reported as the evaluation result.

## Experiments

**Baseline models.** To establish baseline performance, models were first trained from scratch separately on the primary dataset, the ProstateX-2 (PX2) data, and the PROMISE12 (P12) data. Training was performed using five-fold CV over each entire dataset, with 324 images per fold. The evaluation metrics were then computed using the approach described above.

**Generalizability to challenge datasets.** To assess the utility of the baseline primary dataset model on the external challenge datasets, two sets of experiments were done for each dataset. First, the model was used to produce segmentation mask predictions for each example in the external datasets, and mean scores were reported for each dataset. Then, the model was refined for each external dataset using the baseline primary dataset model as the pretrained weight initializer. This refining was done using five-fold CV over 100 epochs, and validation soft Dice scores were calculated and reported as in the previous experiments. Results were compared for superiority against the baseline models using a one-tailed paired $t$-test, with $\alpha = 0.001$.

**Impact of dataset ablation.** To assess the impact of the size of the primary dataset on both segmentation performance and generalizability, a series of ablation experiments were conducted. First, a series of models was trained using truncated versions of the primary dataset. In these experiments, the training set for each fold was truncated to a fixed proportion of its original size, from 5% to 80%. The validation set was not truncated to ensure a fair comparison. Five-fold CV was again used over 100 epochs. The resulting models were then evaluated using the soft Dice criterion to determine model performance on the primary dataset. In order to determine the impact of ablation on generalizability, the resulting models were then refined for 100 epochs using the PX2 or P12 datasets (without truncation), and then evaluated as in the previous experiments. These models were compared for superiority against the baseline models using one-tailed paired $t$-tests, with $\alpha = 0.001$.

**Comparison to BraTS model.** In order to assess the relative importance of using the domain-specific primary baseline model as a pretrained weight initializer, a comparison model was trained using the BraTS dataset. The BraTS dataset was chosen for the comparison model because of the similar underlying data (T2-weighted imaging) and 3D nature of the desired output. The BraTS data was preprocessed using the same pipeline before training as

**Table 1. Evaluation results for baseline models.**

| | Soft Dice Coefficients | | | | Average Hausdorff Distance |
|---|---|---|---|---|---|
| Dataset | Overall | Base | Midgland | Apex | |
| **Primary** | 0.909 ± 0.042 | 0.863 ± 0.095 | 0.941 ± 0.030 | 0.832 ± 0.094 | 0.156 ± 0.231 |
| **PX2** | 0.702 ± 0.083 | 0.679 ± 0.117 | 0.849 ± 0.051 | 0.702 ± 0.093 | 0.480 ± 0.555 |
| **P12** | 0.568 ± 0.122 | 0.501 ± 0.168 | 0.762 ± 0.087 | 0.561 ± 0.168 | 2.155 ± 2.466 |

PX2 = ProstateX-2, P12 = PROMISE12, results reported as mean ± standard deviation across all images

the prostate data, and the same model architecture and training protocol was used. Five-fold CV was performed over 150 epochs. The BraTS model was then used as a pretrained weight initializer for refining PX2 and P12 segmentation models, using the same approach as in the previous experiments. These models were compared for superiority against the baseline models using one-tailed paired $t$-tests, with $\alpha = 0.001$; additionally, the refined ablation models were compared against the refined BraTS models for superiority using one-tailed paired $t$-tests, with $\alpha = 0.001$.

## Results

### Baseline models

Training results are shown in **Table 1**; all results are reported as mean ± standard deviation in tables and text. The primary baseline model achieved a high overall performance, with a mean overall Dice coefficient of 0.909 ± 0.042 and mean AHD of 0.156 ± 0.231. This result is comparable to the top performing prostate segmentation models found in the literature. Example evaluation segmentations for the baseline model on the primary dataset are shown in **Fig 2, S1** and **S2 Figs**. The PX2 and P12 models performed less well, with mean overall Dice coefficients

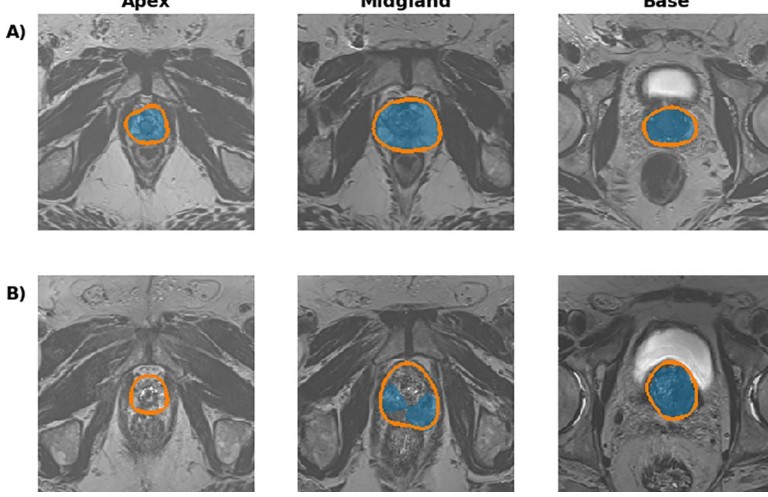

**Fig 2. Example UCLA baseline model segmentations.** The orange contour depicts ground truth segmentation and the shaded blue area depicts model segmentation. **A**) Example apex, midgland, and base slice from a sample in the primary dataset with a high metric on evaluation. The soft Dice coefficient for this sample was 0.928, and the average Hausdorff distance was 0.085. Images of all of the slices for this study are presented in **S1 Fig. B**) Example apex, midgland, and base slice from a sample in the primary dataset with a low metric on evaluation. The soft Dice coefficient for this sample was 0.738, and the average Hausdorff distance was 0.935. Images of all of the slices for this study are presented in **S2 Fig**.

of 0.702 ± 0.083 and 0.568 ± 0.122, and mean AHDs of 0.480 ± 0.555 and 2.155 ± 2.466, respectively. Across all three models, midgland Dice performance was the highest (0.762–0.941) and performance on the base and apex regions was more limited (0.501–0.863). The P12 model was the worst performing across every measure. Performance measures on a per-sample basis for the PX2 and P12 baseline models are shown in Fig 3.

### Generalizability to challenge datasets

Results are shown in Table 2. For the PX2 dataset, the primary baseline model exhibited a mean overall Dice coefficient of 0.465 ± 0.291 and AHD of 4.824 ± 5.920 before refining, and a coefficient of 0.912 ± 0.029 and AHD of 0.150 ± 0.192 after refining. For the P12 dataset, the primary baseline model exhibited an overall Dice coefficient of 0.708 ± 0.210 and AHD of 1.953 ± 3.747 before refining and a Dice of 0.852 ± 0.091 and AHD of 0.581 ± 1.314 after refining. Similar to the previous experiments, Dice performance in the midgland region was higher than that in the base and apex regions for all models. For both datasets, the refined primary baseline model performed significantly better ($p < 0.001$) than the baseline model trained with only the respective dataset across all measures. Though the unrefined UCLA model performed better on the P12 dataset, after refining, performance was best on the PX2 dataset. Performance measures on a per-sample basis for the PX2 and P12 refined baseline models are shown in Fig 3. Example segmentations before and after refining are shown in S3 and S4 Figs.

### Impact of dataset ablation

Results for these experiments are shown in Fig 4 and Table 3. We found that model performance generally increased as the proportion of data used increased, with the primary model exhibiting an overall mean Dice coefficient of 0.638 at 5% and 0.909 at 100%. Both the PX2 and P12 models exhibited significantly increased performance ($p < 0.001$) over their baseline at all ablation levels. For all three sets of models, the models trained at the 60% ablation level achieved approximately the eightieth percentile performance.

### Comparison to BraTS Model

The results of these experiments are shown in Table 4. The final overall soft Dice coefficient of the resulting model on the BraTS segmentation task was 0.591. When refined on the PX2 dataset, the mean overall soft Dice coefficient was 0.834, and the AHD was 0.299 ± 0.465. When refined on the P12 dataset, the mean overall Dice coefficient was 0.704, and the AHD was 1.428 ± 2.603. In both cases, the refined BraTS model significantly outperformed the baseline model across all measures ($p<0.001$), but was outperformed by the ablation models at 20% and higher ($p<0.001$). Performance measures on a per-sample basis for the PX2 and P12 refined BraTS models are shown in Fig 3. Example segmentations are shown in S3 and S4 Figs.

### Discussion

In this study, we developed a prostate segmentation CNN model using a large clinically generated dataset, and examined the relationship between dataset size and model performance. We further explored the generalizability of the model to external datasets, and the relative contribution of using the model as a pre-trained starter for improving performance when training on limited datasets.

We found that the network trained on our institution's dataset did not perform well initially when used on outside data. However, refining the network on the external data using the initial model as a pre-trained starter yielded significantly superior performance to training using

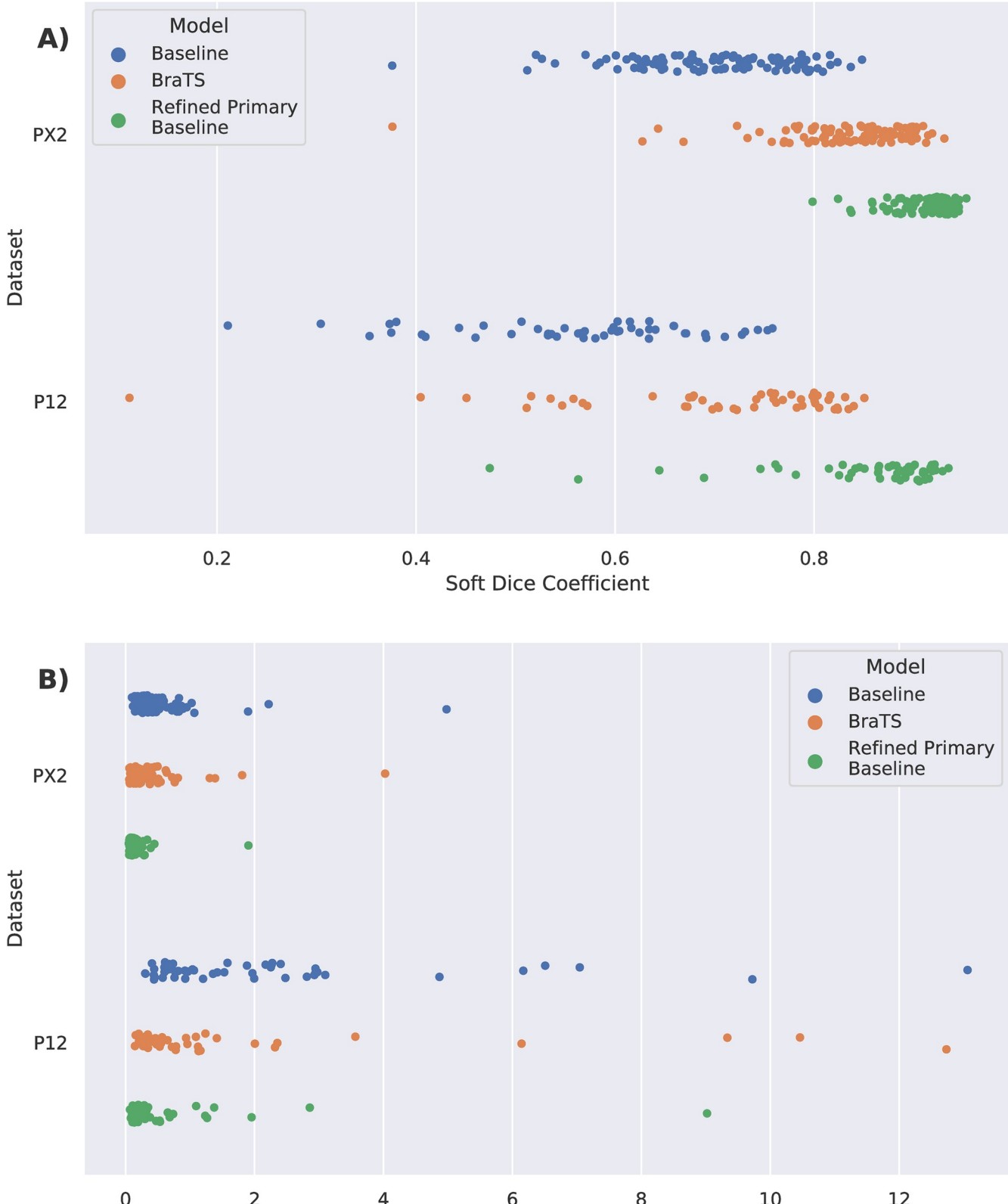

**Fig 3. Evaluation metrics for PX2 and P12 datasets.** Soft Dice coefficients (*A*) and average Hausdorff distances (*B*) for every sample in the ProstateX-2 (PX2, *n* = 99) and PROMISE12 (P12, *n* = 50) datasets, after model evaluation for the baseline, BraTS, and refined primary baseline models. Each solid dot represents a single training example. The models trained by refining the BraTS pretrained model or the baseline pretrained model both exhibited improved performance and reduced variance on both evaluation metrics, and with the refined primary baseline model exhibiting the highest performance and lowest variance. Detailed statistics are available in **Tables 1**, **2**, and **4**.

randomly initialized models. On the PX2 dataset, using our institution's model as a pre-trained starter yielded an increase in mean overall Dice coefficient of 30%, and on the P12 dataset, an increase of 49%. Using a model trained on data (BraTS 2019) completely unrelated to prostate segmentation as a pre-trained starter also yielded improvements over baseline, but was not as effective as using the primary dataset as a starter. As demonstrated in **Fig 3**, model performance improved progressively from the baseline model, to the model trained using the non-relevant BraTS MR data, and finally to the model trained with the highly relevant UCLA prostate MR data. The final performance of the models we trained using our pre-trained prostate MR starter was comparable to other results from the literature on both datasets using more complex models [9, 26, 27], highlighting the value of creating a domain-specific starter for this task. For example, the leading model on the P12 leaderboard (submitted on 9/7/2020) has a Dice score of 0.895, which compares favorably with our final overall Dice coefficient of 0.852 [10]. The leading model trained on a large private dataset (trained on 648 studies at the NIH) has a Dice score of 0.915, which compares favorably with our 0.909 [12].

We also found that using truncated versions of our dataset still yielded significant improvements. Even using a model trained on only 15% of the primary dataset as a pre-trained starter yielded improvements over baseline of 18% and 28% on the PX2 and P12 datasets, and the gains from increasing dataset set saturated at approximately 60% of the primary dataset.

These findings are notable in part because our primary dataset consists of rough clinical contours that have not been carefully re-annotated to produce a machine learning-quality dataset and images that were not filtered for inclusion of only optimal quality series. We included in our primary dataset images with quality limitations, images that used endorectal coils, and images from patients who had had prostate treatments that significantly distort the visual appearance of the prostate. Despite these complications, we still found that we were able to train a state-of-the-art model and then use that model to boost the performance of models trained on "gold-standard" data. The performance gained through the use of our model as a pre-trained starter was greater than that obtained using an unrelated pretrained model (as is typical for transfer learning; i.e. ImageNet [28]), suggesting that our model was able to learn features that were useful starters for the segmentation models trained for the external datasets.

Our work does have some limitations. Because we did not use a machine learning-quality version of our dataset, it is difficult to compare the overall performance results on our data to state-of-the-art models. In addition, the imperfections in the clinically generated ground truth

**Table 2. Evaluation results for retargeted models.**

| Refining? | Dataset | Soft Dice Coefficients | | | | Average Hausdorff Distance |
|---|---|---|---|---|---|---|
| | | Overall | Base | Midgland | Apex | |
| No | PX2 | 0.465 ± 0.291 | 0.314 ± 0.314 | 0.517 ± 0.316 | 0.401 ± 0.312 | 4.824 ± 5.920 |
| Yes | PX2 | 0.912* ± 0.029 | 0.851* ± 0.102 | 0.949* ± 0.024 | 0.849* ± 0.070 | 0.150* ± 0.192 |
| No | P12 | 0.708 ± 0.210 | 0.475 ± 0.317 | 0.779 ± 0.215 | 0.679 ± 0.221 | 1.953 ± 3.747 |
| Yes | P12 | 0.852* ± 0.091 | 0.744* ± 0.207 | 0.918* ± 0.046 | 0.777* ± 0.134 | 0.581* ± 1.314 |

* denotes significantly higher than baseline model, *p*<0.001. PX2 = ProstateX-2, P12 = PROMISE12, results reported as mean ± standard deviation across all images

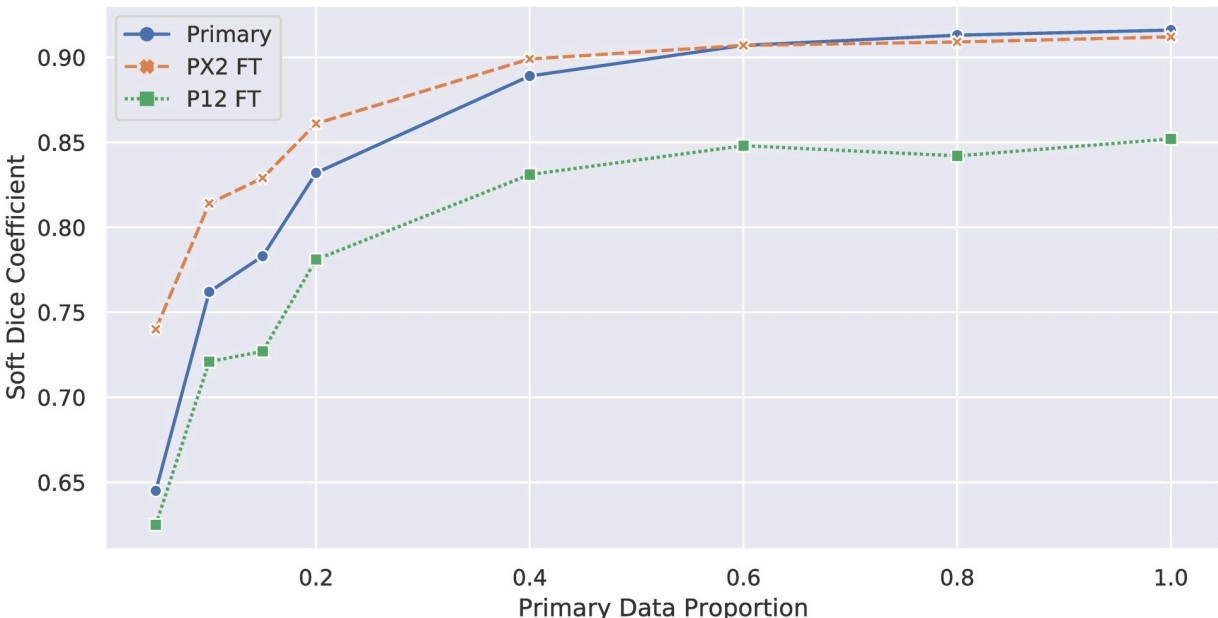

**Fig 4. Soft Dice coefficients for models trained with ablated dataset.** Soft Dice Coefficients for models trained using the ablated primary dataset ("Primary") or trained using an ablated primary model as weight initializer ("FT"). PX2 = ProstateX-2, P12 = PROMISE12, FT = fine-tuned. Significant improvements can be seen in the performance of the fine-tuned models at 5% of the primary dataset used for training the ablated primary baseline model, with the performance benefits leveling out at 60% of the dataset.

segmentations we used for our primary dataset likely include both areas incorrectly annotated in the foreground and the background. As a result, some differences between model predictions and the ground truth in the primary dataset are the result of inaccurate labels, rather than model error.

Because we held the model design constant and simple in order to isolate the dependent variables in our experiments to the datasets used for training and pretraining (and as such used data from all folds in our evaluations, rather than a single held-out fraction), we may have been prevented from realizing performance gains that other works have found through complex model designs or post-processing steps. However, our intent with these choices was to demonstrate that even a simple model with rough clinical contours can provide substantial value when contemplating model development. This finding may have significant implications for future work, in which larger datasets with lower-quality annotations may be combined with smaller datasets with high-quality annotations to maximize the value of available data without requiring the significant expenditure of re-annotation effort. We plan to further explore this hypothesis in future work using more difficult problems, such as prostate cancer segmentation, in order to determine if this approach may unlock additional potential for

**Table 3. Model performance using ablated primary dataset (overall soft Dice coefficients).**

| Model | 5% | 10% | 15% | 20% | 40% | 60% | 80% | 100% |
|---|---|---|---|---|---|---|---|---|
| Primary | 0.638 | 0.754 | 0.775 | 0.825 | 0.883 | 0.901 | 0.906 | 0.909 |
| PX2 FT | 0.740* | 0.814* | 0.829* | 0.861* | 0.899* | 0.907* | 0.909* | 0.912* |
| P12 FT | 0.625* | 0.721* | 0.727* | 0.781* | 0.831* | 0.848* | 0.842* | 0.852* |

* denotes significantly higher than baseline model, $p<0.001$. PX2 = ProstateX-2, P12 = PROMISE12, FT = fine-tuned, results reported as mean across all images

**Table 4. Evaluation results for refined BraTS models.**

| Dataset | Soft Dice Coefficients | | | | Average Hausdorff Distance |
|---|---|---|---|---|---|
| | Overall | Base | Midgland | Apex | |
| **PX2** | 0.834* ± 0.072 | 0.783* ± 0.126 | 0.903* ± 0.065 | 0.775* ± 0.097 | 0.299* ± 0.465 |
| **P12** | 0.704* ± 0.137 | 0.614* ± 0.208 | 0.820* ± 0.120 | 0.644* ± 0.186 | 1.428* ± 2.603 |

* denotes significantly higher than baseline model, $p<0.001$. PX2 = ProstateX-2, P12 = PROMISE12, results reported as mean ± standard deviation across all images

medical image analysis. Additionally, because this is a retrospective analysis and does not include the real-time ultrasound used for image fusion, it is not possible for us to evaluate the impact of segmentation quality from different models on registration and biopsy targeting. Future, prospective work should include such an evaluation.

We trained a state-of-the-art model using rough clinical annotations, producing a prostate segmentation model with a mean overall Dice coefficient of 0.909 and an AHD of 0.156. We additionally found that models trained using truncated fractions of our data were effective pre-trained starters for achieving higher performance models on external prostate segmentation challenge datasets. Our findings suggest a role for the combined use of datasets with low-quality and high-quality annotations in future medical image analysis model development in order to maximize performance while minimizing annotation effort.

## Supporting information

**S1 Table. Imaging acquisition parameters for study datasets.** Full acquisition data is not available for the PROMISE12 dataset, and the counts for images acquired at different field strengths and resolutions are not available.
(DOCX)

**S1 Fig. Full volume example of primary baseline dataset segmentation, high metric.** Orange contour depicts ground truth segmentation. Shaded blue area depicts model segmentation. Slices depicted from apex to base. The soft Dice coefficient for this sample was 0.928, and the average Hausdorff distance was 0.085.
(PNG)

**S2 Fig. Full volume example of primary baseline dataset segmentation, low metric.** Orange contour depicts ground truth segmentation. Shaded blue area depicts model segmentation. Slices depicted from apex to base. The soft Dice coefficient for this sample was 0.738, and the average Hausdorff distance was 0.935.
(PNG)

**S3 Fig. Example ProstateX-2 segmentations.** Orange contour depicts ground truth segmentation. Shaded blue area depicts model segmentation. The soft Dice coefficient and average Hausdorff distance metrics were 0.645 and 1.024 for the baseline model, 0.864 and 0.167 for the BraTS model, and 0.932 and 0.079 for the refined primary baseline model.
(PNG)

**S4 Fig. Example PROMISE12 segmentations.** Orange contour depicts ground truth segmentation. Shaded blue area depicts model segmentation. The soft Dice coefficient and average Hausdorff distance metrics were 0.536 and 2.974 for the baseline model, 0.678 and 0.291 for the BraTS model, and 0.910 and 0.102 for the refined primary baseline model.
(PNG)

## Author Contributions

**Conceptualization:** Karthik V. Sarma, Alex G. Raman, Nikhil J. Dhinagar, Alan M. Priester, Stephanie Harmon, Thomas Sanford, Sherif Mehralivand, Baris Turkbey, Leonard S. Marks, Steven S. Raman, William Speier, Corey W. Arnold.

**Data curation:** Karthik V. Sarma, Alex G. Raman, Nikhil J. Dhinagar, Alan M. Priester, Stephanie Harmon, Thomas Sanford, Sherif Mehralivand, Baris Turkbey, Leonard S. Marks, Steven S. Raman, William Speier.

**Formal analysis:** Karthik V. Sarma, Alex G. Raman, William Speier, Corey W. Arnold.

**Funding acquisition:** Karthik V. Sarma, Leonard S. Marks, William Speier, Corey W. Arnold.

**Investigation:** Karthik V. Sarma, Alex G. Raman.

**Methodology:** Karthik V. Sarma, William Speier, Corey W. Arnold.

**Project administration:** Karthik V. Sarma, Baris Turkbey, William Speier, Corey W. Arnold.

**Resources:** Karthik V. Sarma, Stephanie Harmon, Thomas Sanford, Sherif Mehralivand, Baris Turkbey, Leonard S. Marks, Steven S. Raman, Corey W. Arnold.

**Software:** Karthik V. Sarma, Alex G. Raman, Nikhil J. Dhinagar, Alan M. Priester, Stephanie Harmon.

**Supervision:** Baris Turkbey, Leonard S. Marks, Steven S. Raman, William Speier, Corey W. Arnold.

**Validation:** Karthik V. Sarma.

**Visualization:** Karthik V. Sarma.

**Writing – original draft:** Karthik V. Sarma.

**Writing – review & editing:** Karthik V. Sarma, Alex G. Raman, Nikhil J. Dhinagar, Alan M. Priester, Stephanie Harmon, Thomas Sanford, Sherif Mehralivand, Baris Turkbey, Leonard S. Marks, Steven S. Raman, William Speier, Corey W. Arnold.

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
