## [Decision Letter · Decision Letter 0]

7 Apr 2021

PONE-D-21-06200

Harnessing clinical annotations to improve deep learning performance in prostate segmentation

PLOS ONE

Dear Dr. Sarma,

Thank you for submitting your manuscript to PLOS ONE. After careful consideration, we feel that it has merit but does not fully meet PLOS ONE’s publication criteria as it currently stands. Therefore, we invite you to submit a revised version of the manuscript that addresses the points raised during the review process.

The reviewers have asked for revisions. Some of the reviewers have questioned the novelty of the paper. There is also concerns about the discussion and the comparisons, that authors need to address. Based on all this, I am recommending major revisions. 

Furthermore when submitting the revised paper, please also consider the following points:

1. English language needs proofreading.

2. References should be in proper format.

3. All acronyms must first be defined.

We look forward to receiving your revised manuscript.

Kind regards,

Usman Qamar

Academic Editor

PLOS ONE

Journal Requirements:

Thank you for stating in your Funding Statement:

KVS acknowledges support from National Cancer Institute grant F30CA210329, National Institute of General Medical Studies grant GM08042, and the UCLA-Caltech Medical Scientist Training Program.

CWA acknowledges funding from National Cancer Institute grants R21CA220352 P50CA092131, and an NVIDIA Corporation Academic Hardware Grant.

LSM acknowledges funding from National Cancer Institute grants R01CA195505 and R01CA158627.

SH acknowledges that this project has been funded in whole or in part with federal funds from the National Cancer Institute, National Institutes of Health, under Contract No. HHSN261200800001E. 

TS, SM, and BT acknowledge that this project was supported in part by the Intramural Research Program of the NIH.

The content of this publication does not necessarily reflect the views or policies of the Department of Health and Human Services, nor does mention of trade names, commercial products, or organizations imply endorsement by the US Government. 

Thank you for stating the following in the Competing Interests section:

I have read the journal's policy and the authors of this manuscript have the following competing interests: LSM and AMP report a financial interest in Avenda Health outside the submitted work. BT reports IP-related royalties from Philips. The NIH has cooperative research and development agreements with NVIDIA, Philips, Siemens, Xact Robotics, Celsion Corp, Boston Scientific, and research partnerships with Angiodynamics, ArciTrax, and Exact Imaging. No other authors have competing interests to disclose.

We note that you received funding from a commercial source: Avenda Health , NVIDIA, Philips, Siemens, Xact Robotics, Celsion Corp, Boston Scientific,  Angiodynamics, ArciTrax, and Exact Imaging

We note that you have stated that you will provide repository information for your data at acceptance. Should your manuscript be accepted for publication, we will hold it until you provide the relevant accession numbers or DOIs necessary to access your data. If you wish to make changes to your Data Availability statement, please describe these changes in your cover letter and we will update your Data Availability statement to reflect the information you provide.

Reviewers' comments:

Reviewer's Responses to Questions

**Comments to the Author**

1. Is the manuscript technically sound, and do the data support the conclusions?

Reviewer #1: Yes

2. Has the statistical analysis been performed appropriately and rigorously? 

Reviewer #1: Yes

3. Have the authors made all data underlying the findings in their manuscript fully available?

Reviewer #1: No

4. Is the manuscript presented in an intelligible fashion and written in standard English?

Reviewer #1: Yes

5. Review Comments to the Author

Reviewer #1: The paper addresses an important need in medical image segmentation analysis today, developing accurate segmentation models using unrefined segmentations as inputs. The use of non-perfect inputs can enable more data to be used for training and potentially improve model generalizability.

The authors use a large clinical dataset of >1600 prostate MRIs to develop and validate a deep learning model for prostate gland segmentation. The model uses only clinically-produced prostate segmentations as model inputs. The study appears appropriately designed and utilizes the largest training cohort in prostate segmentation to date. I liked the idea to evaluate the model for a non-prostate segmentation task (i.e., brain cancer segmentation). I have several comments.

Abstract

1. Materials and method: It could be relevant to mention that 5-fold cross-validation was performed, possibly also mentioning the initial train-validation split in the abstract.

2. Results: Please clarify whether the dice score of 0.91 was obtained on the training or independent testing data. It appears that all of the data was used for model training and there was no independent test set on the UCLA data.

Introduction

3. Paragraph 1 sentence 1: I recommend either keeping and using the abbreviation for prostate cancer (PCa) consistently throughout the paper or removing the abbreviation.

4. Paragraph 1 sentence 3: At some institutions, urologists review the segmentations on T2-weighted MRI, whereas other institutions use radiologists or a mix.

5. Paragraph 1 sentence 4: Please change Phillips to Philips. This typo is repeated elsewhere as well.

6. Paragraph 1 sentence 4: If one exists, please provide a citation for DynaCAD Prostate (Philips) segmentation.

7. Paragraph 2 sentence 1: This makes a definitive assumption that inaccurate segmentations are sufficient for clinical tasks. Please provide a citation that supports this argument. Otherwise, I would not state this so definitively. For accurately targeting cancer in the prostate it is not the volume that is most important, but the relationship of the cancer to the edge of the prostate.

8. Paragraph 3: It would enhance the paper if the authors could provide a range in the number of cases that have previously been utilized for training different prostate segmentation models and cite the models; not all prostate segmentation models have been trained on small datasets, e.g.,:

“Data Augmentation and Transfer Learning to Improve Generalizability of an Automated Prostate Segmentation Model. Thomas H. Sanford, Ling Zhang, Stephanie A. Harmon, Jonathan Sackett, Dong Yang, Holger Roth, Ziyue Xu, Deepak Kesani, Sherif Mehralivand, Ronaldo H. Baroni, Tristan Barrett, Rossano Girometti, Aytekin Oto, Andrei S. Purysko, Sheng Xu, Peter A. Pinto, Daguang Xu, Bradford J. Wood, Peter L. Choyke, and Baris Turkbey. American Journal of Roentgenology 2020 215:6, 1403-1410”

Materials and methods

9. Primary dataset: If possible, please mention or describe whether scans were compliant with PIRADS specifications.

10. Primary dataset: To help readers assess generalizability, please describe the distribution between scans obtained on different scanners (e.g., Philips, GE, and/or Siemens).

11. Primary dataset: Please include whether scans were acquired using endorectal coil or not? Did this effect model performance?

12. Model, training and evaluation: Consider calculating the Hausdorff distance in all cases in the internal test set to demonstrate how gland segmentation accuracy may impact the location of the target.

13. Model, training and evaluation: The authors might want to mention that the results were reported as mean +/- standard deviation.

14. Baseline models: What was the initial training-validation split in the 5-fold cross-validation?

15. Please make it clear who performed the expert segmentations used as the gold standard for model training and for calculating dice scores.

Results

16. It appears that the Dice of 0.91 was obtained from the 5 fold cross validation rather than from a held out test set in the UCLA data or from testing in the external dataset. Is this correct? If so, why was testing not performed in a held out test set from the UCLA data?

17. Please report the results as mean +/- standard deviation.

18. The authors mention that the model performs better after refinement. Why is the performance so poor prior to refinement (Dice 0.465)?

19. Is the improvement in dice clinically significant (i.e. does it make the cancer target location more accurate)?

Discussion

20. There appears to be a discrepancy between the results in the abstract and what can be found in the discussion section. Did using the model trained on the large internal dataset as a pre-trained template increase performance (dice score) by 39% and 49% in the two respective external datasets – or was it 30% and 49%?

21. To help put this into context, I’d recommend citing other prostate segmentation papers that perform worse, equal to, and better than your model.

22. In calculating the dice in the clinical dataset, what was used as the gold standard segmentation? If it just the clinical segmentation, is it possible that the model segmentation is better than the gold standard?

Tables and figures:

23. It should be clear to the reader that the results are shown as mean +/- standard deviation.

24. Space permitting, a table that shows relevant MRI characteristics and dataset composition would enhance the paper's value (e.g., it could include the number of cases, scanners, MRI sequence, magnetic field strength, slice thickness, in-plane resolution).

25. There should be figures that demonstrate the performance of the model visually.

26. It would be helpful if the paper provided an overview diagram of the 3D U-Net architecture and possibly also the preprocessing steps in the same diagram.

27. I recommend including 1-2 figures demonstrating representative test cases from the internal and external datasets, showing every slice from base-to-apex, including segmentation outputs before and after refinement.

28. It would be great to see the addition of box plots (or similar plots) that show dice score distributions (range, IQR, median) before and after refinement.

29. It would be interesting to see whether the choice of deep learning model affects the results significantly, more so than the size of the dataset. One idea could be to look at ImageNet or the holistically nested edge detector network (https://www.ncbi.nlm.nih.gov/pmc/articles/PMC5565676)

Limitations section:

30. I recommend incorporating the following into the limitations.

a. The 0.91 dice reported in the abstract is not from a held out test set or the external testing.

6. PLOS authors have the option to publish the peer review history of their article (what does this mean?). If published, this will include your full peer review and any attached files.

Reviewer #1: No

---

## [Author Response · Author response to Decision Letter 0]

16 May 2021

Response to Reviewers

(point-by-point)

Thank you for the very helpful review of our work. Based on your suggestions, we have made numerous improvements to our paper, and have detailed improvements made in response to your comments below:

Abstract

1. Materials and method: It could be relevant to mention that 5-fold cross-validation was performed, possibly also mentioning the initial train-validation split in the abstract.

We have added information about the cross-validation split and clarified that the evaluation scores are calculated across the full dataset to the section.

2. Results: Please clarify whether the dice score of 0.91 was obtained on the training or independent testing data. It appears that all of the data was used for model training and there was no independent test set on the UCLA data.

We have added a parenthetical clarification to the section that the results are across CV folds.

Introduction

3. Paragraph 1 sentence 1: I recommend either keeping and using the abbreviation for prostate cancer (PCa) consistently throughout the paper or removing the abbreviation.

We have removed this abbreviation from the paper for clarity.

4. Paragraph 1 sentence 3: At some institutions, urologists review the segmentations on T2-weighted MRI, whereas other institutions use radiologists or a mix.

We have updated this sentence to more accurately represent the process.

5. Paragraph 1 sentence 4: Please change Phillips to Philips. This typo is repeated elsewhere as well.

We have fixed the spelling of “Philips” throughout the paper.

6. Paragraph 1 sentence 4: If one exists, please provide a citation for DynaCAD Prostate (Philips) segmentation.

Philips has not published any work on the performance or operation of DynaCAD, however we have cited the product website.

7. Paragraph 2 sentence 1: This makes a definitive assumption that inaccurate segmentations are sufficient for clinical tasks. Please provide a citation that supports this argument. Otherwise, I would not state this so definitively. For accurately targeting cancer in the prostate it is not the volume that is most important, but the relationship of the cancer to the edge of the prostate.

We have removed this assertion from the paragraph to put emphasis on the fact that accurate segmentations are critical for downstream image analysis, such as cancer detection.

8. Paragraph 3: It would enhance the paper if the authors could provide a range in the number of cases that have previously been utilized for training different prostate segmentation models and cite the models; not all prostate segmentation models have been trained on small datasets, e.g.,:

“Data Augmentation and Transfer Learning to Improve Generalizability of an Automated Prostate Segmentation Model. Thomas H. Sanford, Ling Zhang, Stephanie A. Harmon, Jonathan Sackett, Dong Yang, Holger Roth, Ziyue Xu, Deepak Kesani, Sherif Mehralivand, Ronaldo H. Baroni, Tristan Barrett, Rossano Girometti, Aytekin Oto, Andrei S. Purysko, Sheng Xu, Peter A. Pinto, Daguang Xu, Bradford J. Wood, Peter L. Choyke, and Baris Turkbey. American Journal of Roentgenology 2020 215:6, 1403-1410”

We have added citations for a sample of previously published prostate segmentation models that use challenge datasets and that use institutional datasets, as well as a range for the sizes of the datasets used.

Materials and methods

9. Primary dataset: If possible, please mention or describe whether scans were compliant with PIRADS specifications.

Our dataset begins in 2010 which predates PI-RADS. As such, we chose to use the T2 SPACE protocol images for our UCLA dataset, as the parameters remained the same through the collection period. We have added additional information about our acquisition protocol and parameters for the T2 images to the Materials and Methods section and table S1; we have also added information about the challenge datasets to table S1.

10. Primary dataset: To help readers assess generalizability, please describe the distribution between scans obtained on different scanners (e.g., Philips, GE, and/or Siemens).

We have clarified in the Materials and Methods section and table S1 that all primary dataset scans were performed on Siemens scanners. 

11. Primary dataset: Please include whether scans were acquired using endorectal coil or not? Did this effect model performance?

An endorectal coil was used for a small fraction (<2%) of patients. We have noted this information in table S1. Because such a small number of patients had an endorectal coil, there is not enough data to conclusively determine the impact of the coil on model performance. For the patients in the primary dataset who were imaged with an endorectal coil, mean overall Dice coefficient was 0.884 (vs. 0.909 across the whole dataset) and mean AHD was 0.200 (vs. 0.156 across the whole dataset); both values are within one SD of the whole dataset means.

12. Model, training and evaluation: Consider calculating the Hausdorff distance in all cases in the internal test set to demonstrate how gland segmentation accuracy may impact the location of the target.

We have added the average Hausdorff distance as an evaluation metric and reported our findings as they relate to model performance improvement throughout the paper, tables, and figures.

13. Model, training and evaluation: The authors might want to mention that the results were reported as mean +/- standard deviation.

We have added additional clarity in this section regarding how the evaluation metrics were computed across all images in the dataset.

14. Baseline models: What was the initial training-validation split in the 5-fold cross-validation?

We have added additional clarity regarding the training/validation set split to the Baseline models section.

15. Please make it clear who performed the expert segmentations used as the gold standard for model training and for calculating dice scores.

We have clarified that the segmentations were performed clinically by the attending radiologist for the case at the time of the initial imaging study; most of the radiologists are not authors of this paper.

Results

16. It appears that the Dice of 0.91 was obtained from the 5 fold cross validation rather than from a held out test set in the UCLA data or from testing in the external dataset. Is this correct? If so, why was testing not performed in a held out test set from the UCLA data?

The mean overall Dice and AHD statistics reported on the primary dataset are the mean over the study-level scores for every image in the dataset included in the evaluation. These scores were calculated using the cross-validation model that was trained without that particular image in the training set (i.e. the image was in the held-out fold for that model). Typically studies require a held-out test set to address the potential bias introduced by model-level optimizations (i.e. adjusting the model design to optimize performance). However, because our study design involved “freezing” the model configuration in order to have the only dependent variables in our experiments be data-related, there was no need for a separate held-out test set because there is no risk of bias introduced from model optimization over the course of the experiments. As such, we chose to use all of the available data in the evaluation in order to provide a fairer assessment of model performance across a larger set of images. We believe this provides the best estimate of model performance and variance.

17. Please report the results as mean +/- standard deviation.

We have updated the paper throughout to report results in this way.

18. The authors mention that the model performs better after refinement. Why is the performance so poor prior to refinement (Dice 0.465)?

The PX2 dataset was acquired with homogeneous imaging parameters, and the P12 dataset was acquired with very heterogeneous imaging parameters. However, the parameter range for the P12 dataset does overlap more closely with the UCLA dataset. We believe this explains why the UCLA model before refining does better on P12 than PX2, and after refining, the PX2 model is superior because of the internal consistency. Unfortunately, because individual sample acquisition parameters were not provided with the P12 dataset, it is not possible to run an analysis to determine whether this hypothesis is correct.

19. Is the improvement in dice clinically significant (i.e. does it make the cancer target location more accurate)?

Unfortunately, it is not possible for us to evaluate the impact of the performance improvements on biopsy targeting due to the retrospective design of our study, which does not include access to the real-time ultrasound used for image fusion. We have added a note to the limitations section of the paper about this and to highlight a prospective evaluation of targeting as future work.

Discussion

20. There appears to be a discrepancy between the results in the abstract and what can be found in the discussion section. Did using the model trained on the large internal dataset as a pre-trained template increase performance (dice score) by 39% and 49% in the two respective external datasets – or was it 30% and 49%?

We have corrected this error (the correct proportions are 30% and 49% improvements in Dice).

21. To help put this into context, I’d recommend citing other prostate segmentation papers that perform worse, equal to, and better than your model.

We have added comparisons and citations to the leading prostate segmentation models on challenge and private datasets to facilitate easier comparison of our results in context to the discussion section of the paper. Since our results are comparable to the best published results available, we provided one comparison for each group in the discussion, and added a series of additional citations for models trained on challenge or private datasets to the introduction section of the paper.

22. In calculating the dice in the clinical dataset, what was used as the gold standard segmentation? If it just the clinical segmentation, is it possible that the model segmentation is better than the gold standard?

For the primary dataset, we used the clinically generated segmentation. As such, it is possible that some differences between model predictions and the ground truth are the result of inaccurate labels, rather than model error. We have added a note to the limitations section about this possibility.

Tables and figures:

23. It should be clear to the reader that the results are shown as mean +/- standard deviation.

We have added this note on each table and to the beginning of the results section text for clarity.

24. Space permitting, a table that shows relevant MRI characteristics and dataset composition would enhance the paper's value (e.g., it could include the number of cases, scanners, MRI sequence, magnetic field strength, slice thickness, in-plane resolution).

We have added this information as table S1.

25. There should be figures that demonstrate the performance of the model visually.

We have added a number of figures (as described below) to help demonstrate our model’s performance and results.

26. It would be helpful if the paper provided an overview diagram of the 3D U-Net architecture and possibly also the preprocessing steps in the same diagram.

We have added such a diagram as Figure 1.

27. I recommend including 1-2 figures demonstrating representative test cases from the internal and external datasets, showing every slice from base-to-apex, including segmentation outputs before and after refinement.

We have added two “every slice” figures demonstrating segmentation outputs as figures S2 and S3. We have also added a figure with representative slices as figure 2. Finally, we have added figures S4 and S5 which depict representative segmentation slices of the same prostate for the baseline, BraTS, and refined primary baseline models.

28. It would be great to see the addition of box plots (or similar plots) that show dice score distributions (range, IQR, median) before and after refinement.

Based on your suggestion, we looked at a number of potential plots (including box plots and violin plots) and found that strip plots best presented the information. We have added strip plots for Dice and average Hausdorff score for the challenge datasets across the baseline and refined models as Figure 3

29. It would be interesting to see whether the choice of deep learning model affects the results significantly, more so than the size of the dataset. One idea could be to look at ImageNet or the holistically nested edge detector network (https://www.ncbi.nlm.nih.gov/pmc/articles/PMC5565676)

We agree that exploring the choice of deep learning model would be of great interest. We chose to fix our model parameters for the purposes of this study in order to focus our analysis on the impact of data size and relevance; however in future work we would like to further study the impact of model design on this outcome. This limitation is noted in the limitations section of the paper.

Limitations section:

30. I recommend incorporating the following into the limitations.

a. The 0.91 dice reported in the abstract is not from a held out test set or the external testing.

We have updated the limitations section to clarify why we held the model design constant and included data from all folds in the valuation, rather than a held-out test set. We have also included performance metrics for the challenge datasets in the abstract as well.

---

## [Decision Letter · Decision Letter 1]

14 Jun 2021

Harnessing clinical annotations to improve deep learning performance in prostate segmentation

PONE-D-21-06200R1

Dear Dr. Sarma,

We’re pleased to inform you that your manuscript has been judged scientifically suitable for publication and will be formally accepted for publication once it meets all outstanding technical requirements.

Kind regards,

Usman Qamar

Academic Editor

PLOS ONE

Additional Editor Comments (optional):

Reviewers' comments:

Reviewer's Responses to Questions

**Comments to the Author**

1. If the authors have adequately addressed your comments raised in a previous round of review and you feel that this manuscript is now acceptable for publication, you may indicate that here to bypass the “Comments to the Author” section, enter your conflict of interest statement in the “Confidential to Editor” section, and submit your "Accept" recommendation.

Reviewer #1: All comments have been addressed

2. Is the manuscript technically sound, and do the data support the conclusions?

Reviewer #1: Yes

3. Has the statistical analysis been performed appropriately and rigorously? 

Reviewer #1: Yes

4. Have the authors made all data underlying the findings in their manuscript fully available?

Reviewer #1: No

5. Is the manuscript presented in an intelligible fashion and written in standard English?

Reviewer #1: Yes

6. Review Comments to the Author

Reviewer #1: The authors have done a nice job responding to my comments. I have no additional recommendations apart from also citing a similar, recent publication on prostate gland segmentation using clinically-generated labels.

Deep Learning Improves Speed and Accuracy of Prostate Gland Segmentations on Magnetic Resonance Imaging for Targeted Biopsy. J Urol. April 2021.

7. PLOS authors have the option to publish the peer review history of their article (what does this mean?). If published, this will include your full peer review and any attached files.

Reviewer #1: No

---

## [Editor Report · Acceptance letter]

18 Jun 2021

PONE-D-21-06200R1 

Harnessing clinical annotations to improve deep learning performance in prostate segmentation 

Dear Dr. Sarma:

I'm pleased to inform you that your manuscript has been deemed suitable for publication in PLOS ONE. Congratulations! Your manuscript is now with our production department. 

Kind regards, 

on behalf of

Dr. Usman Qamar 

Academic Editor

PLOS ONE